# A Multigraph-Based Representation of Hi-C Data

**DOI:** 10.3390/genes13122189

**Published:** 2022-11-23

**Authors:** Diána Makai, András Cseh, Adél Sepsi, Szabolcs Makai

**Affiliations:** 1Department of Biological Resources, Eötvös Loránd Research Network, Centre for Agricultural Research, 2462 Martonvásár, Hungary; 2Department of Molecular Breeding, Eötvös Loránd Research Network, Centre for Agricultural Research, 2462 Martonvásár, Hungary; 3Department of Cereal Breeding, Eötvös Loránd Research Network, Centre for Agricultural Research, 2462 Martonvásár, Hungary

**Keywords:** Hi-C, barley, rice, DNA, graph theory

## Abstract

Chromatin–chromatin interactions and three-dimensional (3D) spatial structures are involved in transcriptional regulation and have a decisive role in DNA replication and repair. To understand how individual genes and their regulatory elements function within the larger genomic context, and how the genome reacts to environmental stimuli, the linear sequence information needs to be interpreted in three-dimensional space, which is still a challenging task. Here, we propose a novel, heuristic approach to represent Hi-C datasets by a whole-genomic pseudo-structure in 3D space. The baseline of our approach is the construction of a multigraph from genomic-sequence data and Hi-C interaction data, then applying a modified force-directed layout algorithm. The resulting layout is a pseudo-structure. While pseudo-structures are not based on direct observation and their details are inherent to settings, surprisingly, they demonstrate interesting, overall similarities of known genome structures of both barley and rice, namely, the Rabl and Rosette-like conformation. It has an exciting potential to be extended by additional omics data (RNA-seq, Chip-seq, etc.), allowing to visualize the dynamics of the pseudo-structures across various tissues or developmental stages. Furthermore, this novel method would make it possible to revisit most Hi-C data accumulated in the public domain in the last decade.

## 1. Introduction

DNA in the living cell is organised into linear chromosomes which undergo dynamic architectural changes during the cell cycle, progressing from elongated interphase chromatin threads to condensed metaphase chromosomes [1]. Within unreplicated chromosomes, a single continuous DNA molecule is tightly packaged into higher order structures. Nucleosomes form the chromatin fibre which are proposed to fold into consecutive chromatin loops [2,3,4] which are assembled into distinct chromosomal domains (topologically associating domains (TADs) in animals [5,6] and TAD-like domain structures in plants [7,8]). The importance of chromatin packaging lays in its impact on DNA accessibility, which consequently affects the fundamental processes of transcription, replication, and recombination. Changes in genome conformation can result in open chromatin regions serving as transcriptional hotspots, while closed genomic regions have been demonstrated to be generally transcriptionally inactive [9,10,11,12,13].

The emergence of the high-throughput chromatin-conformation-capture-sequencing (Hi-C) technologies has greatly contributed to the elucidation of spatial genome organisation in many organisms [8,14,15]. However, determining the structure of the chromatin as it is organised in the cell nucleus is still a challenging task due to its extraordinary complexity and high plasticity. A Hi-C experiment sequences genomic fragments that are spatially co-located. By mapping the paired reads of chromatin capture sequencing to the reference genome, a contact matrix can be computed. Solving the contact matrix yields the putative 3D structure of the genome [16]. This approach represents a powerful tool to map simple, haploid genomes in 3D but appears more challenging when analysing complex, polyploid genomes. Chromatin capture sequencing cannot differentiate between the two parental genomes but chromosome representation can be interpreted as the average of the two homologous chromosomes [17]. 

Hi-C data allows us to make a model of the 3D genome up to a resolution of several thousands of base pairs [18]. In these models, the genome is represented by bins of genomic fragments ranging in length between a few thousand and one million nucleotides. Kremer’s polymer modelling algorithm [19] has been frequently used to model DNA as a bead and springs. The method’s advantage is that it makes the model self-avoiding and computationally feasible. Kremer’s algorithm was originally developed to model polymers where every bead represents a monomer, and the springs represent the bonds between the monomers. Self-avoidance is achieved by setting the diameter of the individual spheres to be greater than the maximum length of a stretched spring. This constraint is reasonable for modelling polymers at a monomer level but may limit the biological relevance of Hi-C inferred 3D-genome models, which are based on bins representing long DNA sequences. 

The DNA is a flexible linear polymer [20,21], which in the living cell nucleus can wind up (like a bath towel) to a fraction of its total length and can unwind and stretch at a scale where transcription, double-strand breaks and recombination occur [22,23]. Genomic bins represent thousands of monomers and the length of the links between these bins exceeds the diameter of the DNA fibre. For that reason, representing bins as monomers connected by short links limits interpretation of the 3D structure. Here, we developed a novel, heuristic approach to represent and analyse large plant genomes in 3D by leveraging tools of graph theory. Our method portrays the genome as a multigraph where Hi-C contact data is mapped to the chromosomes, which are built up from sequentially adjacent bins identified from genomic data. Instead of point-like beads, the bins are represented by ‘elastic’ cylinders which add a new aspect to the representation, notably, that of the lengths of the individual genomic regions (i.e., bins) (Figure 1). Self-avoidance in our modelling is implemented by dynamic checking for the intersection/crossing of cylinders, rather than artificially limiting the length of the links. These elastic cylinders behave like springs and can be stretched following Hooke’s law [24]. 

The motivation of our work is to narrow the gap between big-data biology and direct observations (e.g., microscope). The multigraph representation of the genome and the derived three-dimensional pseudo-structure offers a flexible framework to analyse not only static structures but the dynamics of genome organization in a simple and effective way if data is available. In the present work, our model explored bulk- and single-cell Hi-C data of two major cereal crops, barley (*Hordeum vulgare* cv. Golden Promise, 2n = 2x = 14, HH genome) and rice (*Oryza sativa* ssp. *japonica*, 2n = 2x = 24). We complemented our work with three-dimensional cytological examination of barley chromosomes in cross-linked mitotic interphase nuclei, aided by high-resolution confocal microscopy. 

## 2. Materials and Methods

### 2.1. Raw Data Analyses

The Hi-C library of barley (NCBI Accession No. SRR8922888) used in this study was constructed using DNA extracted from one-week-old seedlings of cv. Golden Promise [25]. The single-cell library of rice (NCBI Accession No. SRR8261290) used here is from sperm isolated from rice stamens [26]. Hi-C data was processed following the method described by Padmarasu and co-workers [27]. We used high-quality ’Golden Promise’ reference assembly [25] and *japonica* reference IRGSP-1.0.

### 2.2. Contact Statistics

Population-averaged (multi-cell) barley library resulted in over 20 million interactions. We used a 10 kb resolution which reduced the interchromosomal contacts to 6.5 M and the intrachromosomal contacts to 5 M. A contact distribution between the 10 kb bins was calculated and a percentile-based threshold of 85 for contact counts was determined to select strong interactions (Figure 2). A strong interaction is characterized by a relatively high number of contacts with very similar frequencies. This is represented by the linear region of the contact distribution diagram (Figure 2a). This number corresponds to percentiles of 0.512 and 0.524 order for intra- and interchromosomal contacts, respectively.

The single-cell library of rice presented fewer contacts. The sperm-cell library had 102,780 interchromosomal and 10.3 M intrachromosomal contacts at 10 kilobase bin size. The data was further filtered based on percentile data. Cutoff values of 11 and 36 for sperm cell were used for inter- and intrachromosomal edges, respectively, for the reasons described above (Figure 2a). Intra-bin contacts (self-loops) were discarded in all cases in our analysis.

### 2.3. Multigraph Construction

Genomes can be perceived as graphs, a network of genomic bins described by a binary adjacency matrix. In this matrix, only adjacent bins have links; all other elements of the matrix are 0. Hi-C analysis also results in an adjacency matrix which is based on captured genomic interactions. In this matrix, the values represent interaction frequencies between the bins (0 means no interaction, any positive integer shows the count of measured interaction). The two graphs (pure contact graph and genomic graph) can be merged and laid out in the three-dimensional space. The resulted construct is a multigraph, as a pair of nodes can have two types of edges. Two adjacent bins are linked by a genomic edge and can (and they very often do) have a contact edge captured by Hi-C data.

In our graph representation of the genomic sequence data, nodes illustrate the starting position of a genomic region, and an edge represents the actual genomic sequence, in other words, the bin. This results in a pure genome graph, and we will refer to these edges as genomic edges (type #1 edge on Figure 1). This representation produces linear graphs equal to the number of chromosomes of the organism under study. Since edges represent the actual chromatin regions that are flexible by nature, their behaviour and spatial layout can be modelled by a spring-model (force-layout) algorithm. Nevertheless, linear graphs where each node has two equal edges will stay roughly linear (note that terminal nodes have a single edge). We used a modified force-layout algorithm to obtain the pseudo-structure. Our layout method applied spatial constrains to approximate the appearance of the genome inside the nucleus. The first constraint was the introduction of a “nucleus”. This was a sphere which limited the spread of “chromosomes” in the space. The subsequent constraint introduced was self-avoidance, which was performed by computing the distances between the edges (segments) during every iteration of the layout. Each time a distance value did meet a threshold value, a repulsive force was applied to the nodes. 

Hi-C data can be interpreted as spatial constrains on the “layout” of the genomic graph by identifying of interacting genomic regions. This constrain was applied by the introduction of new type of edges (type #2 on Figure 1) to the genomic graph and the result is multigraph. The contact edges can have different physical parameters compared to the genomic edges. Our method allows quick experimentation of different parameters (spring coefficient, spring length, etc.). In our case, the contact edges had maximum spring coefficient (hard spring) and ¼ of the spring length of genomic edges. The contact edges were used during the force-layout calculation but excluded from the self-avoidance check.

### 2.4. Calculation of the 2D/3D Layout

The calculation of the 3D layout was based on a physical model of springs [28]. In our method, the forces considered were: (i) repulsion between the edges (self-avoidance), (ii) repulsion between the nodes, (iii) spring force within edges and (iv) drag force of the environment. The present model describes a force-layout algorithm with an addition that (i) selected edges cannot cross, and (ii) the two types of edges (genomic vs. contact) have different spring coefficients and length. 

Here, 3D representation of the genomes (pseudo-structure) was generated by applying the modified force-layout algorithm and hiding the contact edges. Since calculation was relatively fast, we experimented with different settings (not shown). 

Pure contact graphs were modelled by graph analysis as described by Das and co-workers [17]. The 2D modelling of graphs was performed by Gephi [29]. The 3D multigraphs were generated by our own tools using webGL, NodeJS technology and ngraphs [30]. 

### 2.5. Chromosome Territories

The measure of chromosome territories of the pseudo-structures was computed by first calculating the distance matrix of all nodes. Then, we calculated a chromosome level, mean-distance matrix. The diagonal of the matrix shows the mean distances of nodes belonging to the same chromosomes. If these values were the lowest in each column and row, the chromosome’s nodes must be closest to nodes of the same chromosome, thus occupying distinct territories. 

### 2.6. Cytological Analysis

In-situ hybridization was carried out on mitotic root-tip nuclei of barley cv. Golden Promise according to the protocol described by Sepsi and co-workers [31]. The barley centromere-specific G+C sequences [32] and the plant universal telomeric repeat sequences (TRS) were used as probes [33]. 

ImmunoGISH/FISH was performed on mitotic nuclei of a wheat-barley 7BS.7HL disomic translocation line [34] obtained from tapetum cells of the anther tissue. To localise active centromeres, immunolabelling was carried out with a rabbit anti-grass CENH3 antibody [35] detected by a goat anti-rabbit abberior Star Red (STRED-1002-500UG, Abberior Instruments, Göttingen, Germany) secondary antibody. Simultaneous in-situ hybridization included Nick-translation labelled genomic DNA from barley (AF594 NT Labelling Kit, PP- 305L-AF594, Jena Bioscience, Jena, Germany) and the TRS-probe (AF488 NT Labeling Kit, PP-305L-AF488, Jena Bioscience, Jena, Germany).

Confocal microscopy was performed using a TCS SP8 confocal laser-scanning microscope (Leica Microsystems GmbH, Wetzlar, Germany). Z stacks were acquired using a 63X HC PL APO CS2 639/1.40 oil immersion objective (Leica Microsystems GmbH). 

## 3. Results

For both single-cell and bulk (population averaged)-processed Hi-C data, a contact graph was built using bins as nodes and contacts as edges. We added all the bins of each chromosome to the graph, even those that were not participating in any contacts. These nodes only had one or two edges linking them to nodes representing neighboring bins on the chromosome. The resulting graph was a multigraph which contained a graph representation of the full genome integrated with the contact graph. In other words, two types of edges could link a node in the graph, (i) genomic link between nodes representing (sequentially) adjacent bins and (ii) contacts of Hi-C detected interaction which could be either intrachromosomal or interchromosomal (Figure 1). The graph layout algorithm calculated the 3D layout of the multigraphs following the physical model of springs resulting in a three-dimensional Hi-C data and full-genome representation. 

### 3.1. Barley (Multicellular, Bulk Samples)

Population samples can provide rich data sets which are, on the other hand, more difficult to interpret [36]. We used graph analysis tools to obtain a better understanding of the contact data gained from mixed cell types of 10-day-old seedlings of barley. First, we present the pure contact graphs of the barley Hi-C data, and we subsequently introduce the multigraph-based 3D genome model.

Pure contact graphs of barley show conservative contact paths and TAD-like structures.

To reduce complexity, we built contact graphs for interchromosomal and intrachromosomal contacts separately. Both cases demonstrated very complex graphs with non-random layouts (Figure 2b,c and Figure 3a).

The interchromosomal contact graphs showed 73738 bins and 82 807 contacts (Figure 2b). Using the 2D OpenOrd layout algorithm [37], the graph demonstrated a clustered layout with a few bins that were involved in a high number of contacts, while the majority of bins had only contacts to a single bin. We, therefore, filtered nodes by their degree with a cutoff value of 14, which resulted in 371 bins, 297 edges, and a topology of a high number of unlinked subgraphs. We subsequently analyzed the largest subgraph, which demonstrated a chain-like layout of bins (Figure 2c). Given that the pre-filtering (percentile based, see Materials and Methods, contact statistics) selected bins participating in over 85 contacts and the second filtering further filtered these bins for those that participate in at least 14 different contacts, we suggest that the subgraphs represent highly conservative contact paths. Gene enrichment analysis of encoded genes within these bins and in their proximity showed an over-representation of housekeeping genes (Appendix A). Since the experiment was carried out on bulk samples of 10-day-old seedlings, we propose that this network of interacting bins is conserved across all cell types.

When applying the 2D OpenOrd algorithm to bins of a single chromosome with over 85 intrachromosomal contacts, the emerged modular layout displayed topologically distinct node clusters, indicative of TAD-like structures (Figure 3a) [8]. For instance, a 2D graph for the 7H chromosome (Figure 3d) shows nodes at the center of flower-like conformations, further demonstrating that these are TAD-like structures. The colouring of the nodes indicates the position of the bin within the chromosomes. The two terminal segments (telomeres, blue and red) are well-separated and only a few edges are drawn across the blue and red domains, which shows a low number of intrachromosomal contacts between the distal regions detected by the Hi-C analysis. Moreover, on this topology, a third region, located proximally, far from the telomeres (i.e., pericentromeres), is well-separated from the rest of the chromosome (Figure 3a). This characteristic layout, where the distal and proximal regions show a polarised arrangement, depicts the highly conserved Rabl chromosome configuration [38] characteristic of barley interphase chromosomes [39].

We used microscopy techniques to demonstrate the organization of a single barley chromosome arm in mitotic interphase nuclei within a wheat/barley introgression line. The fine structure of the 7H barley chromosome showed chromatin threads as a scaffold, interdigitated with the neighbouring chromatin fibres (Figure 3b). Optical sectioning showed the polarization of the telomeres and centromeres in opposite nuclear hemispheres while the barley 7H chromosome arm pair run perpendicular between the two nuclear poles and occupied separate territories (Figure 3b–d).

### 3.2. Three-Dimensional Layout of the Multigraph Representation of the Genome Reflects Chromosome Territories and a Polarized Configuration

To build the multigraph, we first merged all intrachromosomal (Figure 4a, left) and interchromosomal contact graphs into a combined contact graph (Figure 4a, right). We then added genomic nodes to the combined graph that do not participate in any contacts and edges and that correspond to type #1 edges on Figure 1. This approach allowed the visualization of the part of the genome that was not affected by the combined contact graph. By laying out this multigraph, we were allowed to create a pseudo-genome-structure in three dimensions. To allow easy interpretation of this pseudo-structure, only genomic edges were drawn and checked against intercrossing with other, non-adjacent genomic edges. The contact edges (type #2 edge, Figure 1) that entangle the linear structure of the genome graphs are hidden and were not checked for crossing. 

In the 3D layout, well-defined chromosome territories were observed where the chromosomes clearly folded in half (Figure 4b). Calculating the distance matrix of bins for each chromosome further confirmed the presence of territories (Figure 5a). Every simulation was inherent to the initial state; therefore, the final layout changed as the initial state of the multigraph was altered. However, independent of the initial state, chromosomes always folded in half, implying that the telomeric region of chromosomes clustered together, creating a nucleolus-like, cavernous (hollow) formation spontaneously emerging during the simulation (Figure 6d). This conformation was off center, pushed towards the perimeter of the nucleus depending on the initial changes and/or the modifications of different parameters during the simulation. In cases when this formation was smeared along the nucleus, an architecture reminiscent of the experimentally observed Rabl configuration appeared (Figure 6).

The calculated layout reflected the polarized nuclear organization shown by our microscopic 3D imaging of barley cv. Golden Promise mitotic interphase nuclei (Figure 6a–c), where centromeres and telomeres were polarized in the opposite hemisphere of the nucleus in all cases (Figure 6a–c). The chromatin run perpendicular as parallel threads, enclosing one/two centrally located large nucleolus (Figure 6b).

The pseudo-structure obtained by the multigraph representation thus reflects features resembling nuclear organization observed on optical sections collected from telomere-centromere-labelled 3D nuclei captured by high-resolution microscopy and visualized by 3D rendering (Figure 6b). Mapping expression data along the constructed multigraph model provides a powerful tool which instantly highlights transcriptional hotspots relative to interchromosomal contact frequencies (Figure 4b,c).

### 3.3. Rice (Single Cell Experiments)

We applied our data representation method to haploid single-cell data of rice (*sc*Hi-C datasets) generated by Zhou and co-workers [26]. A haploid sperm-cell library dataset was selected as haploid single-cell data offers unambiguous contact frequencies compared to polyploid bulk samples. However, the large variance in displayed count number was assumed to be the effect of biases during amplification and sequencing [40].

### 3.4. Pure Contact Graphs of Rice

The interchromosomal contact graph of the sperm cell had 1063 nodes and 1861 edges while the intrachromosomal contact graph had 161 nodes and 122 edges after the percentile-based filtering. The merged contact graph had 1125 nodes and 1983 edges. The contact frequency showed a large variance.

### 3.5. Multigraph Representation of Single Cell Hi-C Data of Rice

The intrachromosomal contacts were rather sparse whereas interchromosomal contacts were more abundant for the sperm cells library (Figure 7a). The pseudo-structure displayed a tightly entangled graph of the chromosomes, with distinct chromosomal territories (Figure 5b). Most of the contacts were accumulated in the center of the sphere, and this high concentration of interchromosomal contacts gave rise to a central cavity, which corresponded to the approximate nuclear position of the nucleolus.

### 3.6. Comparative Analysis of Barley and Rice Hi-C Data

The genomes of rice and barley differ in both size and their reported interphase chromosome configuration. Both the 2D and 3D layout of the intrachromosomal multigraphs showed distinct features for barley and rice chromosome organization. The pseudo-structure of individual barley chromosomes was entangled by their two ends and had large loose loops in between (Figure 4 and Figure 6). In the case of rice, chromosome ends appeared to be less interactive as per Hi-C data; however, its pseudo-structure had loops more frequently around a center. This was reminiscent of a flower-like structure (see Appendix A). The pseudo-structures corresponded to the historically reported Rabl- and Rosette-like conformations of barley and rice, respectively. This suggests that multigraph-based pseudo-structures can outline major nuclear topologies and the organization of the chromosomes based on Hi-C data.

## 4. Discussion

Conformation of the genomes in the nucleus is affected by a multitude of constraints and parameters, and the average of all these are inevitably reflected in the chromatin capture data. The output of Hi-C experiments is a list of interactions between genomic regions. Thus, the contact data can be represented as a graph where the nodes are the bins [17].

The herein-introduced multigraph model aims to represent genomes in three-dimensional space by creating a pseudo-structure based on available Hi-C data. The multigraph representation offers a flexible structure which can be extended by additional omics data (RNA-seq, Chip-seq, etc.), allowing to visualize dynamics of the pseudo-structures across various tissues or developmental stages. The efficient integration of this “multiomics” data could help to extract information from interaction networks and find hidden patterns from omics data. A limitation of the work is that it cannot be scaled beyond the scale of Hi-C capture. In addition, the current framework does not model the thickness of chromatin, it only indicates how much force is present in the springs representing genomic regions. Finally, pseudo-structures are not based on direct observation and heuristics to initial settings; however they can be used as a visual aid for the relative positioning of genomic regions.

The chromatin 3D structure is assumed to be determined by its function and not by its thermodynamic equilibrium [41,42]. Elements of higher order chromatin organization and chromatin dynamics show considerable flexibility during the cell cycle. Specific features of chromatin architecture may vary from species to species and cell to cell while conserved phenomena exist. For instance, in plant nuclei, interphase chromosomes are organized in distinct territories [43,44] where landmark chromosomal regions have a well-defined orientation. A highly conserved aspect of chromatin organisation in large genome cereals, such as wheat and barley, is the in a so-called “Rabl” configuration (orientation of their telomeres and centromeres into opposite poles of the nuclear periphery) [39,45]. Rice nuclei, in contrast, carrying the smallest genome among the grasses [46], show a “Rosette-like” configuration, where pericentromeric repeat elements are located in the proximity of the nucleolus from which coding regions emerge as chromatin loops [45,47,48]. Both types of configuration have their characteristic contact pattern.

Interestingly, the contact-graph analysis of rice indicated a conformation similar to that earlier reported as the KNOT conformation by Grob and co-workers [8]. In barley, the filtering of the contact graph highlighted a single contact pathway that resembles Newton’s cradle (Figure 2). Such a linear contact path may hint at evidence of the mechanistic principles governing the genome [49,50]. We examined the genes that are typically abundant within these locations and found over-representation of house-keeping (HK) genes. While these results are in-line with expectations, gene ontology (GO) enrichment analysis tends to highlight HK genes, as they are the most accurately (thus, most frequently) annotated genes [51]. Nevertheless, it is reasonable to think that a subset of bins that form a highly conserved interaction network across cell types connect genes with conserved functions. This further elucidates that these bins may be constitutively transcriptionally active and bear a decondensed, open chromatin structure.

In the pseudo-structure, most of the interchromosomal contacts gathered around a naturally occurring cavity, resembling the nucleolus in living cells. This might suggest that activity shaping the genome is present at the nucleolus perimeter and that the Hi-C technique may be particularly sensitive to contacts around the nucleolus. However, further Hi-C analysis and comparison is needed to confirm this hypothesis. The formaldehyde treatment used by the Hi-C technology to crosslink hydrogen bonds may equally capture temporarily collocated regions, such as enhancer–promoter interactions and more stable connections, such as euchromatin regions localized in the proximity of the nuclear envelope or the nucleolus [52].

Another compelling insight was offered by the fact that our method allowed the observation of chromosome dynamics. During in-silico experiments, we tried to simulate the untangling of the pseudo chromatids by setting the length of the genomic edges shorter (as if in a condensed state) and turning off the effect of contact edges. We noted that certain pseudo-conformations become locked up and cannot be untangled during simulated metaphases, when chromosomes need to separate and migrate to opposite poles of the cell. While our pseudo-structure cannot consider all parameters acting on the DNA molecules in vivo, and the available data is scarce, we allowed edges representing genomic regions to break. This condition alone facilitated the untangling of all pseudo-conformations tested by our experiments. Our observations, thus, show the effectiveness of the technique in highlighting the effect of, e.g., DNA breaks, on the topological complications caused by the processing of the DNA double helix during the cell cycle. Thus, the combination of current Hi-C technologies with mathematical modelling and pseudo-modelling would enable study of the evolutionary advantage of key DNA events, e.g., the effect of DNA breaks and repair mechanisms on cell-cycle control and acceleration.

## Figures and Tables

**Figure 1 genes-13-02189-f001:**
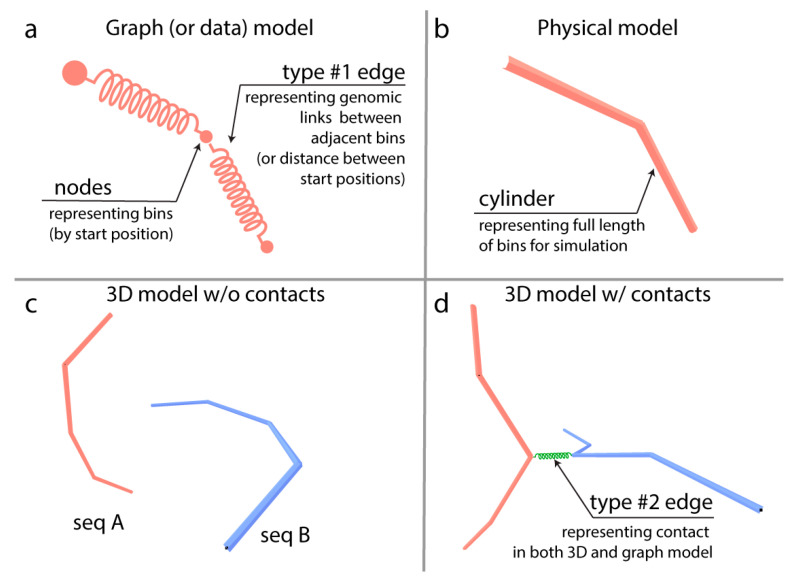
A visual summary of the multigraph model. (**a**) The graph/data model depicts the genome as a graph of adjacent bins. (**b**) The physical interpretation of the graph-based genome model is that nodes represent the start positions of the bins and edges are the physical representation of the bins which are visualized as cylinders. (**c**) Two dummy chromosomes (seq A and seq B) without contacts set in an energy-minimalizing configuration. By adjusting the properties of the bins, the cylinders can be shorter or longer which, in turn, can be used as a model for condensed or decondensed state. (**d**) Between nodes, a type #2 edge can be introduced based on contact data from Hi-C experiments. This puts extra strains on bins, therefore changing the configuration and length of cylinders (while its parameters are unchanged); hence, a graph-based 3D genome. Since the genome and the contact graphs are two different graphs of the same node set (called vertices in graph theory), our 3D genome model is a multigraph.

**Figure 2 genes-13-02189-f002:**
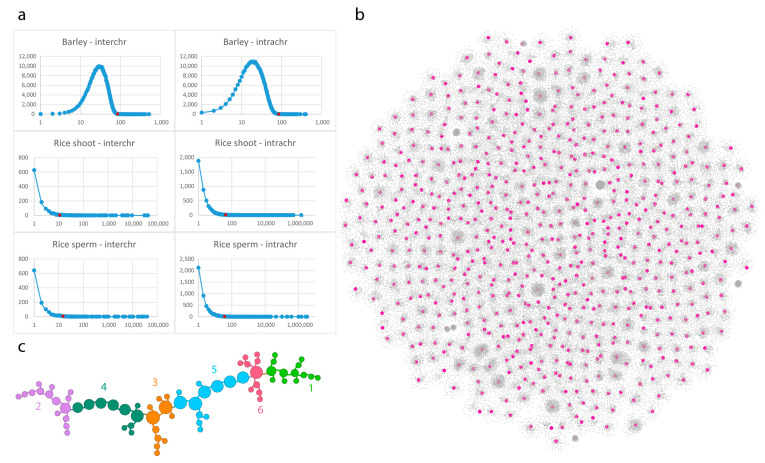
Contact graph construction and analysis. (**a**) Contacts per bin distribution of selected Hi-C libraries. The curve shows the number of bins that have a given contact count. The *x*-axis is logarithmic, and shows contact count; the *y*-axis is linear and shows the number of bins. On the charts, a red dot shows the cut-off value for each analysis (see numbers in the Section 2 of the manuscript). For further downstream analysis, only bins that have a frequency of this value or higher were used. In all cases, self-loops were discarded and not counted. (**b**) A 2D representation of interchromosomal filtered contacts of barley (See methods). Magenta nodes represent bins that have 14, or more than 14, edges in the graph. The layout indicates that contacts between the non-homologuous chromosomes are organized by dedicated genomic regions, suggesting hierarchical genomic structures. (**c**) The largest subgraph after filtering for bins of 14 or more different contacts. An intriguing linear layout of contacts has emerged, suggesting a signal transduction “highway” that might function like Newton’s cradle. Colors represent the clusters of the graph. The clusters were characterized by gene enrichment analysis which revealed housekeeping genes within the regions represented by the bins.

**Figure 3 genes-13-02189-f003:**
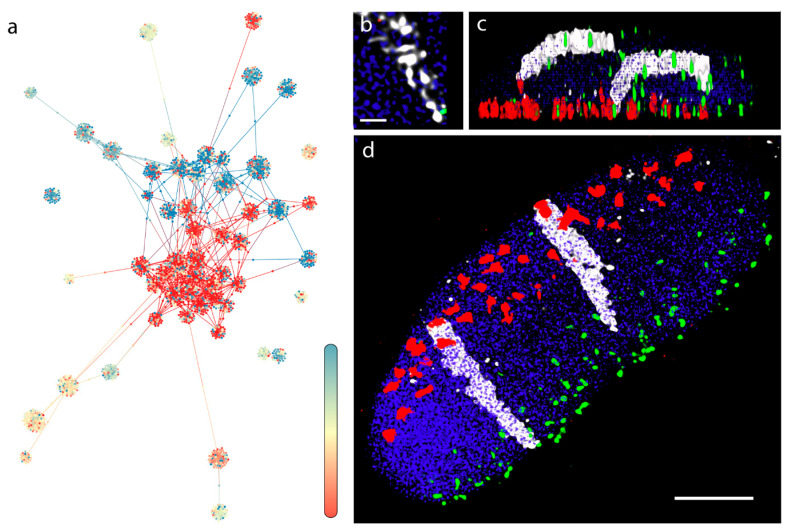
2D graph representation of the contact data of chromosome 7H of barley and microscopic evaluation of the organization of an individual (long) arm of the 7H barley chromosome (7HL) within the interphase nucleus. (**a**) A 2D graph representation of contact data of chromosome 7H. Contacts are filtered for intrachromosomal and 85 hits or above. Fully saturated blue and red colors represent the two telomeric ends of the chromosome. Positions are indicated by a gradient color in between the ends, where pale yellow marks the putative location of the pericentromeric region. Contacts are clustered within each telomeric regions with a few contacts between them. The pericentromeric region is well-separated from the telomeres, a contact pattern typical of the Rabl configuration. (**b**) A high-resolution image enlargement of the 7HL chromatin threads (white) intertwined with the neighboring chromatin captured within a mitotic nucleus of the 7BS-7HL wheat-barley translocation line with confocal laser-scanning microscopy. The 7BS-7HL wheat-barley translocation line carries the 7HL barley chromosome arms stably transferred into the wheat background, allowing the 3D visualization of the barley chromatin by genomic in-situ hybridization (GISH). (**c**) Side view of a three-dimensionally rendered z-stack captured from a mitotic nucleus of the 7BS-7HL wheat-barley translocation line. The 7HL barley chromatin is detected by GISH (shown in white), centromeres are visualized by simultaneous immunolabelling (red) and telomeres are shown by fluorescence in-situ hybridization (green). (**d**) Front view of the same three-dimensionally rendered z-stack, showing parallel arrangement of the two homologous 7H long arms. Rabl organization is clear from the polarized centromere (red signals)-telomere (green signals) arrangement and the signal of the perpendicularly running barley chromosome arms. Bar = 5 μm, except Figure 3b where bar represents 1 μm.

**Figure 4 genes-13-02189-f004:**
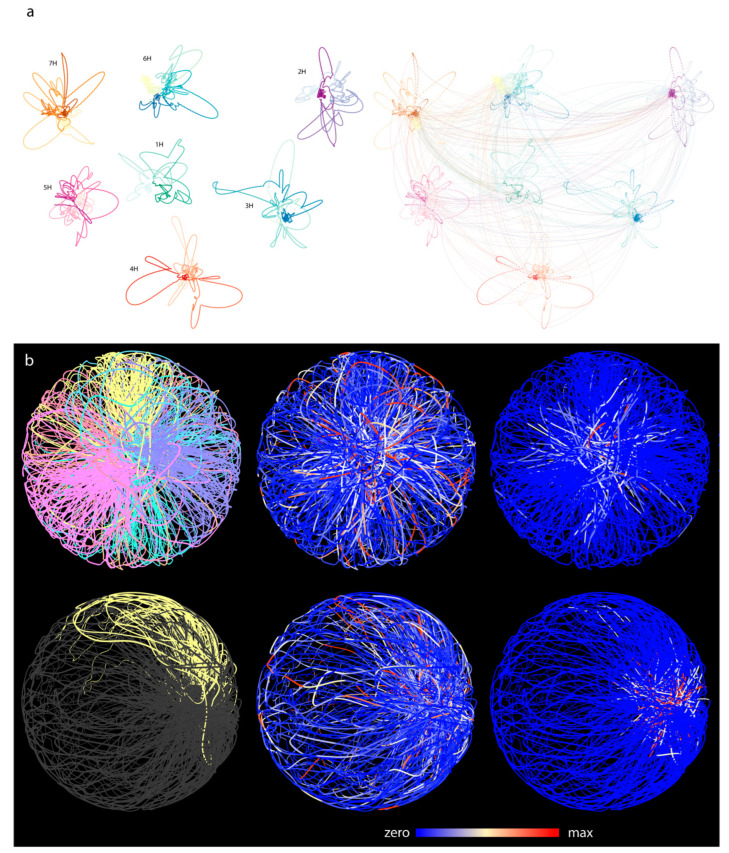
Going from 2D contact graph to multigraph model of 3D genome of barley. (**a**) Two-dimensional layouts of the multigraph of each barley chromosomes. Layout was calculated based on the intrachromosomal contacts only. Entanglement of the two ends of the chromosomes is clearly visible indicating the Rabl configuration. Right panel: interchromosomal contacts are added. (**b**) The three-dimensional pseudo-structure of barley genome. The two rows represent two orthogonal perspectives of the pseudo-structure. Coloring is based on different attributes. On the left panels, chromosomes are color coded (top) and the 5H chromosomes highlighted (bottom). On the central images, expression libraries of various tissues are mapped to the bins (source: the PRJEB14349 bioproject). On the right panels, colors indicate the frequency of interchromosomal contacts. Telomeric regions are enriched in highly conserved interchromosomal contacts.

**Figure 5 genes-13-02189-f005:**
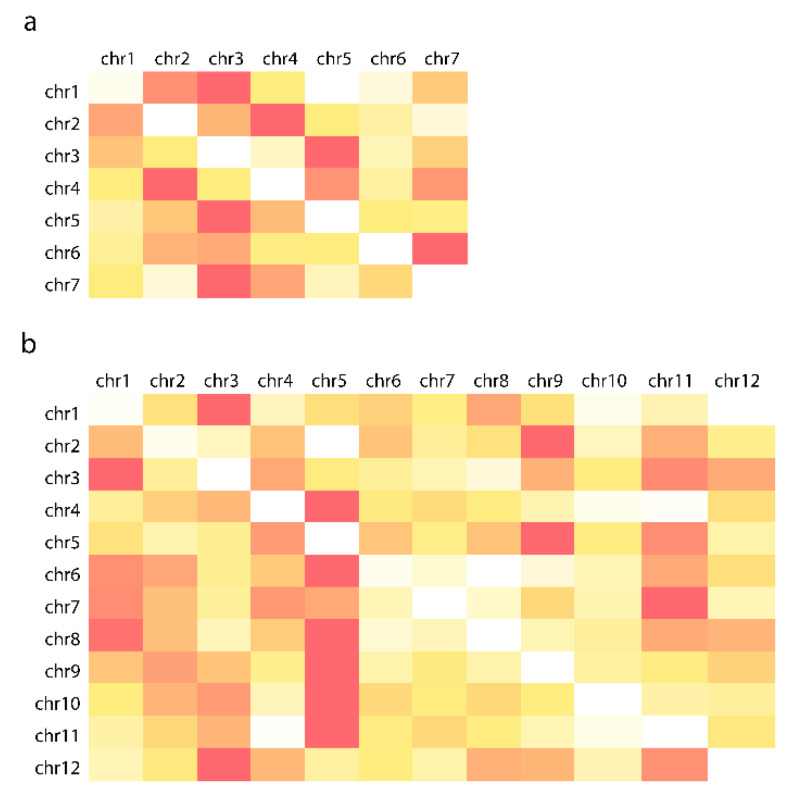
Mean distances of bins grouped by chromosomes. The diagonal white rectangles indicate that in the pseudo-structure of barley (**a**) and rice (**b**), bins of the same chromosomes are closer to each other compared to bins located on other chromosomes. The darker the shade the larger is the value, white is the minimum of the row.

**Figure 6 genes-13-02189-f006:**
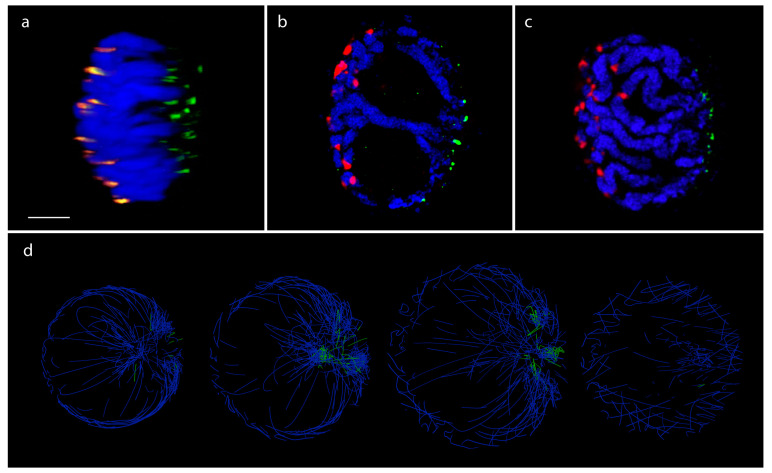
In situ and in silico. Fluorescence in-situ hybridization (FISH) showing centromeres (orange, red) and telomeres (green) on a 3D-fixed barley mitotic nucleus. Barley centromeres are labelled by the barley centromere-specific G+C repeat probe (red) while telomeres are visualized by the plant telomeric repeat sequences (TRS) used as a probe (green). Chromosomes are counterstained with DAPI. Bar = 5 μm. (**a**) Side view of a 3D-rendered z-stack captured from a barley mitotic nucleus. (**b**,**c**) FISH on individual image frames of a barley mitotic nucleus showing polarized centromere-telomere arrangement and chromosome arms laying parallel with the telomere-centromere axis. (**d**) In silico, cross-section images of the pseudo-structure of Barley genome. Green indicates telomeric regions. A well-distinguished nucleolus-like region is visible inside the spherical topology. Telomeres are clustered on one side of the spherical model.

**Figure 7 genes-13-02189-f007:**
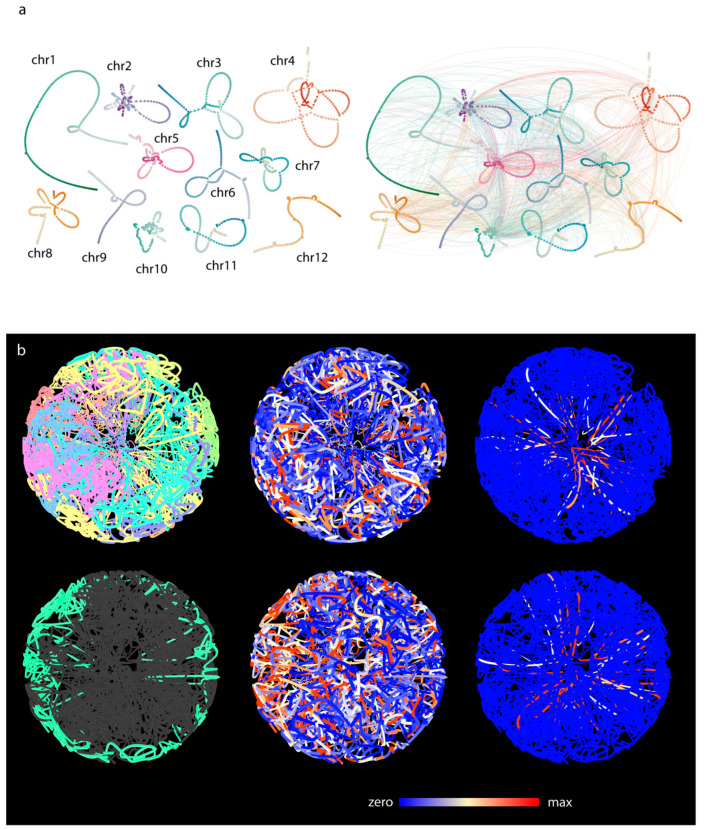
Rice *sc*Hi-C data-based genome graph construction. (**a**) The 2D multigraph image of rice sperm cell. Left without interchromosomal contacts, right interchromosomal contacts drawn. (**b**) Three-dimensional pseudo-structure of rice sperm cell. The two rows represent two orthogonal perspectives of the pseudo-structure. Coloring is based on different attributes. Chromosome color coding shows apparent chromosome territories (top left). Green highlight indicates chr5 (bottom left). On the central images, expression library of sperm cells (GEO accession number: GSE50777) is mapped to the bins.

## Data Availability

The datasets generated and analyzed during the current study are available from the corresponding author upon reasonable request.

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
