# Peer review of "A Multigraph-Based Representation of Hi-C Data"

_genes, 2022, doi:10.3390/genes13122189_

Round 1
Reviewer 1 Report
Manuscript entitled “A multigraph-based representation of Hi-C data” showing an approach to represent HiC datasets in 3D genome. It’s an interesting job. And I have few comments on this paper.
1, In line 84, “2n=2x=24” should not use italics.
2, In line 84, “Oryza sativa ssp. Japonica ” should be “Oryza sativa ssp. japonica”. Rice have two sub-species: japonica and indica.
3, In line 102, “No. SRR8922888” have more than one space.
Author Response
Response to Reviewer 1:
Thank you very much for the encouraging review. We appreciate your points and corrected all the remarks that you highlighted for us.
1, In line 84, “2n=2x=24” should not use italics.
Response 1: We changed the formatting.
2, In line 84, “Oryza sativa ssp. Japonica ” should be “Oryza sativa ssp. japonica”. Rice have two sub-species: japonica and indica.
Response 2: We used the proper naming conventions as suggested.
3, In line 102, “No. SRR8922888” have more than one space.
Response 3: We corrected the typo.
Reviewer 2 Report
In this study, authors proposed a novel approach to represent Hi-C datasets by a whole-genomic pseudo-structure in 3D space, explored two major cereal crops, barley and rice.
Main points:
Too much background information in the Abstract, more key results should be presented.
In the introduction section, according to my point of view, the aims of this study should be added.
Limitations of the work must be mentioned in the discussion section.
Minor points:
Line 132, “highway”.
Line 234, error! Reference source not found?
Line 469-472, duplicate references.
Line 479-560, many references are missing page number information.
Author Response
Response to Reviewer 2:
Thank you very much for the encouraging review. We feel your comments added to the value of the manuscript. Please find our replies below:
- Too much background information in the Abstract, more key results should be presented.
Response 1: We cut some of the sentences covering background info and expanded on results and perspective of the result.
- In the introduction section, according to my point of view, the aims of this study should be added.
Response 2: We elaborated on the aims of the study, and added a new paragraph:
The motivation of our work is to narrow the gap between big data biology and direct observations (eg. microspcope). The multigraph representation of the genome and the derived three-dimensional pseudo-structure offers a flexible framework to analyse not only static structures but the dynamics of genome organization in a simple and effective way if data is available.
- Limitations of the work must be mentioned in the discussion section.
Response 3: We added new sentences to describe the limitation of the work:
A limitation of the work, that it cannot be scaled beyond the scale of Hi-C capture. In addition, the current framework is not modelling the thickness of chromatin, only indicates how much force is present in the springs representing genomic regions. Finally, pseudo-structures are not based on direct observation and heuristic to initial settings, however it can be used as a visual aid for the relative positioning of genomic regions.
- Minor points
Response 4: we corrected the typos, removed the duplicate reference and provided the missing page numbers. In addition, we also formatted the references as per the journals requirements.